# The Relationship between the Use of Non-Verbal Information in Communication and Student Connectedness and Engagement in Online Design Education

Xiaolin Zhang [1], Suyang Cao [2] and Younghuan Pan [3,*]

1 College of Art and Design, Guangdong University of Technology, Guangzhou 510000, China
2 Department of Network Information, Jiangsu College of Safety Technology, Xuzhou 221000, China
3 Interaction Design Lab, Kookmin University, Seoul 01706, Republic of Korea
* Correspondence: peterpan@kookmin.ac.kr; Tel.: +82-2-910-4943

**Abstract:** The COVID-19 pandemic poses a threat to the sustainability of higher education. Connectedness and engagement, two characteristics crucial to design education, have weakened significantly in online courses. However, limited research has been conducted on online design education than on STEM fields. Based on the dual coding theory, the purpose of this study was to use non-verbal tools to enhance design student connectedness and engagement in online class communication. In a quasi-experiment, 122 design students from a Chinese university were questioned and analyzed. They were randomly assigned to four different teaching situations and the effectiveness of two non-verbal tools was tested: emoticons and shared whiteboards. The ANOVA revealed a positive correlation between the use of non-verbal information in online class communication and the connectedness and engagement of design students. Moreover, the students in the group who used plentiful personified-form emoticons gave feedback and reported a stronger sense of connectedness and engagement. The whiteboard group's data did not significantly differ from the control group, unlike the STEM discipline. To better develop the sustainability of design education, we provide recommendations for the design of online-education software and the method of online design instruction.

**Keywords:** connectedness; engagement; online design education; higher-education sustainability; non-verbal cognition

## 1. Introduction

Following the international COVID-19 pandemic, sustainability is becoming more important in education, which poses a challenge not only to the education community but also to society as a whole. Online education has solidified its position as a sustainable method [1]. Educators and students have had to rapidly change their teaching methods and learning strategies to adapt to the software and online learning environment [2,3]. Notwithstanding that students come to understand how to learn online, challenges related to the practical application [4]. For a teaching and learning process to be successful, it must be dynamic and cooperative. If online education is to be more effective, its communication experience needs to be enhanced. This shift in pedagogy fundamentally also challenges many disciplines and disciplinary norms, particularly in the disciplines of art and design. Current online synchronous learning environments still present many barriers to classroom communication, such as access to timely feedback from teachers and peers, loss of information through facial expressions and body language, and technical issues with online software [5,6]. Students' creativity, ability to explore and sense of connectedness and engagement are diminished [7].

If the sustainability of online education is to be enhanced, its communication experience needs to be improved [8,9]. Implementation of teaching communication is closely

linked to the teaching software platform used. The synchronous teaching software currently in widespread use around the world, such as Zoom and Tencent Meetings, was not developed for teaching purposes but to be used as a virtual meeting room by both industry and teaching institutions. Defining, segmenting, and using interactive features for the social needs of classroom activities is an important issue for software designers and educators [10]. In reality, some designers and educators have become interested in how knowledge is transmitted through learning platforms and software features [11]. However, there is relatively little research on how designing and testing these tools can influence student connection and engagement to facilitate online classroom communication. In terms of subject classification, more online education research has focused on the teaching and learning of STEM subjects (Science, Technology, Engineering, and Mathematics) [12]. Numerous online learning systems have also been designed to be more applicable to STEM-related teaching and learning activities [13]. However, the pedagogy and objectives of design disciplines differ significantly from STEM disciplines. The design students' behavioral and cognitive characteristics during communication are inadequately understood. In this article, we will first review the characteristics of online design education activities and the importance of communication in an online environment. Second, design students' cognitive characteristics will be analyzed in terms of how they contribute to communication. Thirdly and finally, quasi-experimental teaching will examine and discuss the relationship between the use of non-verbal information and students' connectedness and engagement.

## 2. Literature Review

### 2.1. Main Design Education Method and Characteristics

The purpose of design education is to enable students to develop creativity, expressiveness, and flexibility, and to permit them to express their inner emotions [14]. Group discussion is an efficient teaching method for design studies. Students discuss and debate a variety of topics that can be investigated in greater detail as the course progresses [15]. Additionally, group work is a common learning method in design education. When students enter the design field, they are frequently required to work in teams. Students who have participated in collaborative projects will be better prepared for the workforce. Furthermore, these students will eventually comprehend and respect the complexities of each design profession. The disadvantage of group work is the possibility of members providing lengthy and laborious explanations, which wastes time and frequently slows down the progress of the project [16–19]. On the other hand, research indicates that high rates of assimilation activities (reading, watching videos, and listening to audio) negatively correlate with student achievement [20]. Therefore, it is necessary to comprehend what promotes an effective and constructive dialogue between students.

### 2.2. Synchronous Online Education

Online education is a broad term that encompasses a variety of instructional environments and approaches [21,22]. There are two basic ways to make learning available to students online and these are called synchronous and asynchronous. The benefits of a web-based approach are the ability to break down spatial constraints and hyperlink to more detailed information in order to support the sustainability of higher education when needed [10]. However, current online education environments still present many barriers to classroom communication, such as access to timely feedback from teachers and peers, loss of information through facial expressions and body language, and technical issues with online software [5,6]. Students' creativity, ability to explore and sense of connectedness and engagement are diminished [7].

The importance of interactions in an online course has been supported by research studies from various theoretical perspectives. The key to successful online education is instructor feedback. Smooth communication activities are associated with a higher percentage of student success [20]. The importance of interaction in identity construction

and learning cannot be overstated [23]. Students' feelings of depersonalization and isolation have a negative impact on student persistence [24].

### 2.3. Connectedness and Engagement

Students' feelings of coherence, essence, beliefs, and interconnectivity are referred to as connectedness [25–27]. Connectedness is the feeling of belonging and acceptance and the creation of bonding relationships [27]. In addition to deciding which teaching activities will be used, choosing the right software tool is critical in developing connectedness. The amount and quality of students' mental, abstract reasoning, sentimental, and reactions to the education process and classroom activities that lead to successful learning outcomes are defined as student engagement. The three components of student engagement in the classroom are cognitive engagement, emotional engagement, and behavioral engagement [28]. Student connectedness and engagement are positively correlated with interactive experiences and are important indicators of effective online learning, as well as predictors of grades and course completion. From all of these findings, increasing connectedness and engagement is essential to developing successful online learning communication.

### 2.4. Dual Coding Theory and Online Environmental Factors

The dual coding theory proposes that the human mind is split into two separate subsystems that process both nonverbal and linguistic information [29]. They can activate from one typifying unit to another thanks to two separate systems. Furthermore, obtaining information codes from both subsystems at the same time can improve educational functionality. The mixture of visual and auditory information, for example, has improved learning ability. Humans have separate information processing channels for visual representation materials and auditory representation materials. When receiving information such as animations or illustrations, the visual channel processes it; when receiving narrative or other sound information, the auditory channel processes it. Engaging all channels in educational activities can be more effective in promoting learning [30–32]. In offline educational communication, students receive linguistic information from teachers and peers through hearing, and non-verbal information from teachers and peers through expressions, body language and learning materials.

When teaching offline, all the information channels are integrated. However, the online education system splits them into different functions. The linguistic information required for interactive communication is achieved through the live call function. The non-verbal information is split into functions such as cameras, emoticons and whiteboards. We define online environmental factors as the necessary functional and interactive elements of the online platform that can support learning. To accomplish beneficial and effective online education, learners must be supplied with a positive learning atmosphere through the use of devices [33]. In online communication, there is little loss of linguistic information conveyed due to environmental factors. The available web technology and voice capabilities ensure that each classroom participant can hear the other members and converse in real-time. However, the opposite may be the case for the communication of non-verbal information. Some students are wary of using cameras in online classrooms [34]. Part of the subjective reason is the students' reluctance to reveal their privacy or to be monitored. Further objective reasons include network speed limitations, particularly for some economically underdeveloped areas, a large number of participants with cameras on at the same time may cause network lag. The lack of visual connectedness resulted in the classroom speakers not always knowing whether others were listening or what their attitudes were towards the ideas being presented. Finally, a student's inspiration may suffer as a result of brief contact with instructors and colleagues [35].

### 3. The Focus of the Present Study

In this study, we began by outlining some key definitions that we found particularly useful. Studies have shown that improving the interactive experience of synchronous online

courses requires compensating for the loss of non-verbal information. Features other than cameras that provide non-verbal information are underutilized in online environmental factors (e.g., emoticons, whiteboards) and they may offer the possibility of addressing the problem of lost non-verbal information.

We decided to perform research to answer the following questions:

**RQ1**. *How do design majors perceive connectedness and engagement with instructors and classmates in synchronous online courses?*

**RQ2**. *How do emoticons and shared whiteboard usage impact students' connectedness and engagement?*

Emoticons are employed in nonverbal communication to emphasize emotional manifestations and meanings [36]. The intricacy and concreteness of emoticons have an impact on emoticon users' performance. In studies of their form and status factors, various emoticons may display distinctions in usefulness [37]. Several research findings have combined abstract and shapes properties into a single category. Moreover, emoticons contain a plethora of personified forms [38]. During synchronous online courses, researchers had seen a requirement to concentrate on graphic emoticons and their expressive form and status.

We propose the following hypothesis:

**Hypothesis 1a (H1a).** *The use of abstract/geometric form emoticons in a synchronous online education environment reports higher levels of perceived connectedness and engagement.*

**Hypothesis 1b (H1b).** *Plentiful personified-form emoticon usage in a synchronous online education environment reports higher levels of perceived connectedness and engagement.*

Multiple studies have found that whiteboard-based instruction enhances cognitive results by increasing motivation and excitement [39,40], enhancing engagement [41–43], improving academic performance, and promoting self-efficacy and learning interest. A shared whiteboard was incorporated into synchronous courses, and the use of a digital whiteboard aided in the understanding of abstract concepts and increased class engagement [44,45]. When compared to traditional lecture-based classes, whiteboard-based instruction can improve students' cognitive learning outcomes [46,47]. Whereas limited research has been conducted to facilitate communication in synchronous online educational environments by using the shared whiteboard.

We hypothesize that:

**Hypothesis 2 (H2).** *Shared whiteboard usage in a synchronous online education environment reports a higher level of perceived connectedness and engagement.*

## 4. Methods

This study took place among design undergraduate students at a Chinese institution. The questionnaire was translated into Chinese and subjected to expert consultation and pilot testing to determine its applicability before being formally distributed. The participants were determined using the convenience sampling method. The quasi-teaching experiment was conducted in the fall semester of 2021. Data collection and analysis of variance took place in the spring of 2022.

### 4.1. Participants

For the main survey, the participants were 122 undergraduate design students from a state university in China. They were invited to complete an online survey about their course's online learning experience; the thirty students who answered the pilot test were precluded. The participants were third-year digital media design and animation students who were studying in China during the COVID-19 quarantine period. All students partici-

pating in the project received an invitation to complete an online survey and to respond anonymously. The final sample contained 65 (53.28%) females and 57 (46.72%) males. The age of participants ranged from 20 to 22 years old, with an average age of 21. All participants have at least one semester of experience in an online course.

*4.2. Procedures*

We measured all constructs using previously validated scales [25–28]. The 22-item instrument measured two sub-scales, connectedness and learning engagement. The survey used a five-point Likert scale, with responses ranging from strongly disagree to strongly agree. With a maximum score of 110 and a minimum score of 22, the total score was computed by summing the points allocated to each of the five-point elements. Ten of the statements were "connectedness items" taken from the Classroom Community Scale [26,48,49]. Higher scores represent a stronger sense of connectedness, and lower scores represent a weaker sense of connectedness. The connectedness sub-scale "represents the feelings of the community of students regarding their connectedness, cohesion, spirit, trust, and interdependence" [27]. Participants' online learning engagement was measured by the learning engagement sub-scale [28,50,51]. The sub-scale contains 12 items.

Firstly, the questionnaire was sent to ten professors at various public universities, and a pilot test with 30 participants was undertaken to identify instrument concerns such as phrasing, substance, and ambiguity. Minor adjustments to the survey were made as a result of their comments. All of the items listed above were included in the questionnaire. Three items in the connectedness sub-scale loaded weakly onto the factor, according to confirmatory factor analyses. As a result, the connection sub-scale was reduced to a seven-item factor. All items loaded reliably (>0.7) onto their appropriate factors after these adjustments (see Appendix A).

The study employed a between-group quasi-experimental approach to evaluate the effectiveness of emoticons, shared whiteboards, and just online teaching. The perceptions of the participants' connectedness and engagement were the outcome variables. Otherwise, the instructor, course content, course length, and course evaluation remained constant. Randomized controlled trials ensure that any differences in average outcomes between the treatment and control groups are due to 'treatment' rather than uncontrollable external causes [52]. Four learning groups were created for comparison in the current study's randomized trials (pure online group, abstract/geometric form emoticons usage, plentiful personified form emoticons usage, and shared whiteboard usage). The following are the teaching approaches for each group. A video creation course was provided to all students via an online tutorial. The coursework was conducted in groups of 2–4 people, with each project group receiving 15 min of free speaking time. Instructor and learners expressed their opinions on project creation and amended plans during class discussion. Tencent conference software was used for all of our courses. In the pure online group (P-O), the instructor only used live video meetings to communicate with the students. For the abstract/geometric form emoticons group (A-E), during the communication process, the instructor used abstract/geometric form emoticons to communicate feelings or give feedback at the same time as the video session. For the plentiful personified-form emoticons group (P-E), the instructor employed the same quantity and semantics of emoticons as the A-E group, but the visual design of emoticons was more humanistic and figurative (see Table 1). For the shared whiteboard group (S-W), the instructor and students communicated using shared whiteboards to convey ideas in the form of sketches.

**Table 1.** The emoticons used in quasi-teaching experiments.

| Emotions | "Hello" | "Good" | "Fighting" | Ask Questions | Encourage |
|---|---|---|---|---|---|
| the abstract/geometric form |  |  |  |  |  |
| the plentiful personified form |  |  |  |  |  |

The demographics of the final samples are shown in Table 2. Because all participants were randomly assigned to each test group, there was no need to control for participant demographics [53].

**Table 2.** Demographics of participants in each group.

| Group | Male | Female | Total |
|---|---|---|---|
| Pure online (P-O) | 16 | 14 | 30 |
| Abstract/geometric form emoticons (A-E) | 15 | 15 | 30 |
| Plentiful personified-form emoticons (P-E) | 13 | 18 | 31 |
| Shared whiteboard (S-W) | 15 | 16 | 31 |

After all of the groups had completed their teaching experiments, all students were contacted and asked to participate in an anonymous survey. The survey was hosted on the online survey platform. A web link to the survey was sent by email. Students were told that the results of the survey were not related to their course grades. The measurements were obtained for further comparison.

## 5. Data Analysis and Finding

The validity and reliability of the measurements were assessed once the questionnaire data were collected. To examine reliability on the total and sub-scales, connectedness and engagement, Cronbach's alpha analysis and correlation analysis were used. The total and sub-scale Cronbach's alpha levels were 0.92, 0.72 and 0.93, respectively, indicating that each had an adequate level of inter-item reliability. Each group underwent a descriptive statistical analysis. Levene's statistic was used to test the premise of homogeneity of variances. Each construct reliability (CR) exceeded 0.7. Additionally, each average variance extracted (AVE) exceeded 0.5. The factors' descriptive, reliability, and validity data are shown in Table 3.

**Table 3.** Reliability and validity analysis.

| Variable | Number of Items | M | SD | AVE | CR | Cronbach's Alpha |
|---|---|---|---|---|---|---|
| Connectedness | 7 | 26.279 | 4.268 | 0.546 | 0.792 | 0.72 |
| Engagement | 12 | 49.771 | 7.012 | 0.580 | 0.942 | 0.93 |
| Total | 19 | 76.049 | 10.452 | | | 0.92 |

A one-way ANOVA was then used to test for differences between each learning method. To examine the differences between the potential pairs of means, we used the Tukey HSD test [54]. To put it another way, if the one-way ANOVA shows that not all means are equal, the Tukey HSD tells which pairs of means are not equal. This strategy eliminates methodological errors and enables precise repeated comparisons [55]. Table 4 shows the results of the one-way

ANOVA. In terms of total (F = 6.504, $p < 0.001$), connectedness (F = 5.325, $p < 0.01$), and engagement (F = 5.929, $p < 0.01$), the four groups are found to be significantly different. This means that students in each group have varying degrees of connectedness and engagement. A greater level of significance is reported for connectedness.

**Table 4.** Results of one-way ANOVA test.

| Variable | Comparison | Sum of Squares | df | Mean Square | F | Sig. |
|---|---|---|---|---|---|---|
| Total | Between Groups | 1875.396 | 3 | 625.132 | 6.504 *** | 0.000 |
| | Within Groups | 11,342.309 | 118 | 96.121 | | |
| Connectedness | Between Groups | 262.874 | 3 | 87.625 | 5.325 * | 0.002 |
| | Within Groups | 1941.651 | 118 | 16.455 | | |
| Engagement | Between Groups | 779.331 | 3 | 259.777 | 5.929 ** | 0.001 |
| | Within Groups | 5170.243 | 118 | 43.813 | | |

Note. * $p < 0.05$, ** $p < 0.01$, *** $p < 0.001$.

Table 5 below shows a more detailed comparison of the four datasets. The use of plentiful personified form emoticons was more active in convincing design students' connectedness than pure online learning (mean difference = 3.98817, $p = 0.05$) and abstract/geometric form emoticons (mean difference = 2.75, $p = 0.05$), but not significantly different from the shared whiteboard (mean difference = 1.65, $p = 0.384$). Design students' connectedness in the pure online group was not significantly different from the abstract/geometric form emoticons group (mean difference = $-1.23$, $p = 0.642$) or the shared whiteboard group (mean difference = $-2.34$, $p = 0.115$), according to the findings. The engagement of the plentiful personified form emoticons group was significantly higher than the pure online group (mean difference = 7.09, $p < 0.001$), but not significantly different from all the other groups.

**Table 5.** Results of post hoc tests (Tukey HSD).

| Variable | Comparison | MD | Std. Error | Sig. | 95% Confidence Interval | |
|---|---|---|---|---|---|---|
| | | | | | Lower Bound | Upper Bound |
| Connectedness | P-O vs. A-E | −1.2333 | 1.0474 | 0.642 | −3.9628 | 1.4961 |
| | P-O vs. P-E | −3.9882 ** | 1.0389 | 0.001 | −6.6955 | −1.2808 |
| | P-O vs. S-W | −2.3430 | 1.0389 | 0.115 | −5.0504 | 0.3643 |
| | A-E vs. P-E | −2.7548 * | 1.0389 | 0.044 | −5.4622 | −0.0475 |
| | A-E vs. S-W | −1.1097 | 1.0389 | 0.710 | −3.8170 | 1.5977 |
| | P-E vs. S-W | 1.6452 | 1.0303 | 0.384 | −1.0399 | 4.3302 |
| Engagement | P-O vs. A-E | −3.9333 | 1.7091 | 0.104 | −8.3873 | 0.5206 |
| | P-O vs. P-E | −7.0893 *** | 1.6953 | 0.000 | −11.5071 | −2.6713 |
| | P-O vs. S-W | −3.0247 | 1.6953 | 0.286 | −7.4426 | 1.3932 |
| | A-E vs. P-E | −3.1559 | 1.6953 | 0.250 | −7.5738 | 1.2620 |
| | A-E vs. S-W | 0.9086 | 1.6953 | 0.950 | −3.5093 | 5.3265 |
| | P-E vs. S-W | 4.0645 | 1.6813 | 0.079 | −0.3170 | 8.4461 |

Note: (1) P-O represents pure online learning, A-E represents abstract/geometric form emoticons, P-E represents plentiful personified-form emoticons, and S-W represents shared whiteboard; (2) * $p < 0.05$, ** $p < 0.01$, *** $p < 0.001$.

## 6. Discussion

### 6.1. The Use of Emoticons in a Synchronous Online Education Environment Report Higher Levels of Perceived Connectedness and Engagement

The statistical evidence suggests that the method increased student engagement and connectedness overall. Specifically, we found that the use of plentiful personified-form emoticons was significantly related to students' online learning connectedness and engagement. The study's findings agree with the majority of previous research on the effectiveness of emoticons in forecasting student online education communication [56–60].

The above analyses add to the literature on the application of emoticons in online design courses and allow us to investigate the effects of non-verbal information more thoroughly.

*6.2. Shared Whiteboard Usage in a Synchronous Online Education Environment does Not Report a Higher Level of Perceived Connectedness and Engagement*

Unlike the results of related studies in other disciplines, the connectedness and engagement of the shared whiteboard group were not statistically different from other groups [61]. In retrospect, one probable explanation is that, unlike STEM subjects, design studies students used the common whiteboard for more than just discussion. The utilization of a shared whiteboard by students from other disciplines is a visual process. However, for design students, drawing is a fundamental skill. The shared whiteboard is only used for sharing and illustrating ideas and the quality of improvised drawings is not an indicator for teachers to evaluate students. Yet, the psychological pressure of demonstrating skills still causes many students to avoid using the whiteboard to produce drafts. In particular, when a student with excellent drawing skills uses the whiteboard first, other students with lesser drawing skills try to avoid it and turn to verbal forms of communication. Another possible reason is that design students are accustomed to using specialist drawing software. Even if they are only used to visualize ideas, the whiteboard functions provided by the online learning system are difficult for them to use because the functionality is too simple and poor in terms of usability and ease of use (e.g., the delay in brushes).

## 7. Limitations and Future Research

First, all participants were chosen from one Chinese four-year university. The findings of this study could be influenced by the university's and student body's features. This study has to be replicated to be generalized beyond the sample size.

Furthermore, we need to better attend to design and experiment with the emotion function's ease of use. The usability of the software functionality was not considered in this study. Easy-to-use interfaces are more inclined to stimulate online communication among instructors and students, resulting in improved online education effectiveness. In future research, the findings from this study will be used as a basis for the design and development of teaching software in conjunction with interaction design theories such as usability and ease of use, and to verify its sustainability in teaching.

## 8. Conclusions

The nature of online design education activities poses a challenge to synchronous online-teaching sustainability. This paper offers insights into improving the quality of online classroom communication from a cognitive perspective, based on the dual coding theory. We assessed the relationship between the use of non-verbal information and design students' perceptions of connectedness and engagement in a synchronous online course through a quasi-teaching experiment. The research involved a quantitative exploration of the role of emoticons and shared whiteboards. Our findings suggest that design students report the most desirable perceptions of connectedness and engagement outcomes in online courses using plentiful personified-form emoticons. Students may benefit from the nonverbal features of the synchronous online education software, thereby increasing their contact with their classmates and instructors. The findings of this study provide key recommendations for the sustainable development of higher education, the organization of teaching by design educators and the development of online-teaching software by interactive designers.

**Author Contributions:** Conceptualization, X.Z. and S.C.; methodology, X.Z.; software, X.Z.; validation, X.Z. and S.C.; formal analysis, X.Z.; investigation, X.Z.; resources, X.Z.; data curation, X.Z.; writing—original draft preparation, X.Z.; writing—review and editing, X.Z.; visualization, X.Z.; supervision, Y.P. All authors have read and agreed to the published version of the manuscript.

**Funding:** This research was funded by the Chinese Ministry of Education Collaborative Education Project between Universities and Firms, grant number 202102025002 and the Guangdong University of Technology Online Course Construction Project, grant number 211210102.

**Institutional Review Board Statement:** The study was conducted in accordance with the Declaration of Helsinki, and approved by the Ethics Committee of Guangdong University of Technology (protocol code GDUT-ART-2020-039 and date of approval 1 September 2020).

**Informed Consent Statement:** Informed consent was obtained from all subjects involved in the study.

**Data Availability Statement:** Not applicable.

**Conflicts of Interest:** The authors declare no conflict of interest.

## Appendix A

**Table A1.** Questionnaire questions.

| Scales | Items |
|---|---|
| Connectedness | 1. I feel that students in this course care about each other.<br>2. I feel connected to others in this course.<br>3. I do not feel a spirit of community.<br>4. I feel that this course is like a family.<br>5. I trust others in this course.<br>6. I feel that members of this course depend on me.<br>7. I feel that I can rely on others in this course. |
| Engagement | 1. I try to do my best during classes.<br>2. I discuss what I have learned in class with my friends out of class.<br>3. My teachers are always near me when I need them.<br>4. I give importance to studying together with my classmates (in a group).<br>5. I have teachers that I can share my problems with.<br>6. I feel like myself as a part/member of a student group.<br>7. I like communicating with my teachers.<br>8. I like seeing my friends in class.<br>9. I am an active student in class.<br>10. I attend classes willingly.<br>11. I carefully listen to my teacher in class.<br>12. I carefully listen to other students in class. |

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
