# Peer review of "The Relationship between the Use of Non-Verbal Information in Communication and Student Connectedness and Engagement in Online Design Education"

_sustainability, doi:10.3390/su142315741_

Round 1
Reviewer 1 Report
This paper uses quasi-experiments to investigate whether the using of online visual teaching tools help to enhance students’ connectedness and engagement in online class. The paper is in general well-written and well-structured. The hypothesizes are clearly stated and the experiments are well designed.
Some minor issues in the text should be improved further, for example,
P2 line 65, Main design education method?
P2 line 66 what the ‘situ’ mean?
Author Response
Point 1: P2 line 65, Main design education method?
Response 1: We are so grateful for your kind question. The logic and language of lines 72 to 86 have been reorganized for presentational clarity. We hope that the revised formulation clarifies our perspectives. The details are as follows:
“The purpose of design education is to enable students to develop creativity, expressiveness, and flexibility, and to permit them to express their inner emotions [14]. Group discussion is an efficient teaching method for design studies. Students discuss and debate a variety of topics that can be investigated in greater detail as the course progresses [15]. Additionally, group work is a common learning method in design education. When students enter the design field, they are frequently required to work in teams. Students who have participated in collaborative projects will be better prepared for the workforce. Furthermore, these students will eventually comprehend and respect the complexities of each design profession. The disadvantage of group work is the possibility of members providing lengthy and laborious explanations, which wastes time and frequently slows down the progress of the project [16-19]. On the other hand, research indicates that high rates of assimilation activities (reading, watching videos, and listening to audio) negatively correlate with student achievement [20]. Therefore, it is necessary to comprehend what promotes an effective and constructive dialogue between students.”
Point 2: P2 line 66 what the ‘situ’ mean?
Response 2: Please accept our sincerest apologies for the simple spelling error. As stated above, the entire paragraph has been rewritten. Again, apologies.

Reviewer 2 Report
The research seems up-to-date and interesting in terms of its subject. Although the research title is not too long, it is suitable for the purpose.
It was also good that the abstract of this research was in the form of a structured summary. In other words, it would be good to write a summary that includes the purpose, method, data collection tool, and summary of the results. However, the abbreviation STEM was used in the abstract and no explanation was written in the introduction. Terminological words such as Visual Information, Engagement, and Student Connectedness are written in the title. However, conceptual details are not given in the introduction. Therefore, the introduction of the study is not appropriate in terms of literature. The introduction part of the research is not sufficient in terms of the subject area. The bibliographies used are up-to-date. Therefore, the use of new bibliography in the introduction and discussion sections of the research has enriched the research.
Purpose and sub-objectives were written in line with the findings. The research method was not written. Survey research method was used in the study. But it must be written. Care was taken to write the tables used in the research in the form of APA6 standard. The data collection tool has not been validated. It could be done in the form of a validity factor analysis. Besides, even content or face validity was not written. There is no problem with reliability. Parametric tests can be performed if the data are normally distributed. However, there is no information on this subject in the study.
The discussion, conclusion, and recommendations section of the research was written as a single section. It would be better if these sections were written separately.
Author Response
Point 1: The abbreviation STEM was used in the abstract and no explanation was written in the introduction.
Response 1: Thank you so much for your comments and professional advice. Based on your suggestion and request, we have added an explanation of STEM in lines 57-62, along with relevant references. The details are as follows:
“In terms of subject classification, more online education research has focused on the teaching and learning of STEM subjects (Science, Technology, Engineering, and Mathematics) [12]. Numerous online learning systems have also been designed to be more applicable to STEM-related teaching and learning activities [13]. However, the pedagogy and objectives of design disciplines differ significantly from STEM disciplines.”
Point 2: Terminological words such as Visual Information, Engagement, and Student Connectedness are written in the title. However, conceptual details are not given in the introduction.
Response 2: To eliminate misinterpretations and ambiguities surrounding the term "Visual Information." The term “Visual Information” has been replaced with “non-verbal information” in the text. From lines 116 to 133, the theoretical origins of the concept and their relevance to this study are described. The details are as follows:
“The dual coding theory proposes that the human mind is split into two separate subsystems that process both nonverbal and linguistic information [29]. They can activate from one typifying unit to another thanks to two separate systems. Furthermore, obtaining information codes from both subsystems at the same time can improve educational functionality. The mixture of visual and auditory information, for example, has improved learning ability. Humans have separate information processing channels for visual representation materials and auditory representation materials. When receiving information such as animations or illustrations, the visual channel processes it; when receiving narrative or other sound information, the auditory channel processes it. Engaging all channels in educational activities can be more effective in promoting learning [30-32]. In offline educational communication, students receive linguistic information from teachers and peers through hearing, and non-verbal information from teachers and peers through expressions, body language and learning materials.
When teaching offline, all the information channels are integrated. However, the online education system splits them into different functions. The linguistic information required for interactive communication is achieved through the live call function. The non-verbal information is split into functions such as cameras, emoticons and whiteboards.”
We have revised lines 106-113 to make the concepts of connectedness and engagement more transparent. The details are as follows:
“Connectedness is the feeling of belonging and acceptance and the creation of bonding relationships [27]. In addition to deciding which teaching activities will be used, choosing the right software tool is critical in developing connectedness. The amount and quality of students' mental, abstract reasoning, sentimental, and reactions to the education process and classroom activities that lead to successful learning outcomes are defined as student engagement. The three components of student engagement in the classroom are cognitive engagement, emotional engagement, and behavioral engagement [28].”
Point 3: The research method was not written.
Response 3: This research used a quasi-experimental research design varying different non-verbal aids. 122 students were split into four groups for a semester-long experiment in teaching. At the conclusion of the course, the experiment's outcomes were evaluated using a questionnaire. In lines 191 through 206, specific details of the research methodology are provided.
Point 4: Care was taken to write the tables used in the research in the form of APA6 standard.
Response 4: All tables in the article have been modified in accordance with the APA6 format.
Point 5: The data collection tool has not been validated. It could be done in the form of a validity factor analysis. Besides, even content or face validity was not written. There is no problem with reliability. Parametric tests can be performed if the data are normally distributed. However, there is no information on this subject in the study.
Response 5: We have added a validity analysis to lines 270-274 of the article. The details are as follows:
“ Each construct reliability (CR) exceed 0.7. And each average variance extracted (AVE) exceed 0.5.”
Point 6: The discussion, conclusion, and recommendations section of the research was written as a single section. It would be better if these sections were written separately.
Response 6: Part five has been rewritten and divided into two sections.: “Data analysis and finding” and “Discussion”.

Reviewer 3 Report
The authors of the paper The Relationship between the Use of Visual Information in Communication and Student Connectedness and Engagement in Online Design Education were inspired by the well-known fact that the communication in virtual environment is reduced due to the lack of nonverbal keys. In online learning context some of the relevant information might be lost and meanings might me misinterpret. It becomes the sustainability issue in the midst of Covid 19 pandemic when it became the only way of learning and teaching.
Although there are lot of common problems of online learning for all kind of subjects, and they are mentioned in the paper, the authors tackle the specific case of design and art students based on creativity but also on production and necessity for continuous product delivering and receiving feedback. For example, the fact that the platform used for educational purposes is not intended for educational but rather business communication and meetings purposes might be a barrier. It means that important aspects of students` cognitive and behavioral characteristics are not taken into consideration and it might alter the efficacy of learning through platform. The question that is raised in the research considers how these disadvantages might be overcome. Authors used quasi-experimental research design varying different visual aids.
They rely on concepts of connectedness and engagement and dual code theoretical framework while defining the hypotheses and providing the explanation of the results. Variations of the quasi-experimental situations are achieved with introducing different visual forms (geometric, personified) and tools (shared white boards) and consequent variables (learning engagement, connectedness) are measured by valid and reliable instruments.
The quality of the paper is in clear and logical research design. Nevertheless, the argumentation why exactly these visual stimuli are chosen here, should be more explicitly stated, and elaborated. There are other alternative tools that can be used beside the shared white board. Also, I am not sure if, strictly theoretical but also practically speaking, emoticons as visual aids and white board as a tool are at the same level of engaging and enriching online communication. That is one of the reasons, I suppose, why shared white board did not prove to be the advantage and to make difference comparing the control group (beside the mentioned explanation).
Also, I would recommend further elaboration of the issue and deeper theoretically explained results but I understand why authors keep narrative in this level of depth (more at the surface) . They describe the results they came up with but for future research it should go more profoundly to the question of the role of visual elements to supersede limitations of online learning.
So, I would like authors to offer some shortly explained reason they used these kind of stimuli (I understand abstract and personified emoticons as visual tools but how shared white board is in the same line with other two tools). Intuitively, I can understand why they use it but theoretically and in practice, they might have different implications (shared white board is more behavioral and more engaging tool, I suppose).
Also, in conclusion I would recommend more referring to theoretical framework.
Author Response
Point 1: I would like authors to offer some shortly explained reason they used these kind of stimuli (I understand abstract and personified emoticons as visual tools but how shared white board is in the same line with other two tools).
Response 1: We gratefully appreciate your valuable comments. And we totally understand your concern. The two tools validated in this paper are two of the most widely discussed and widely used non-verbal tools in online practical courses. However, the majority of past research evidence has come from STEM disciplines, and the purpose of this paper is to determine if they are equally applicable to design disciplines. As the reviewer states, there are indeed other non-verbal tools available, but due to experimental time and conditions, we have only validated these two tools at this stage. According to the reviewer's suggestion, we will report the results of the validation of the other tools in the future, and we have already begun working on this. Regarding the whiteboard, it is true that it is rarely used to convey emotion and sentiment in other disciplines. In design, however, the whiteboard is utilized primarily for drawing, which is a visual form of communicating emotion. Thanks again to the reviewer for the comments, which will also be helpful for our future research.

Round 2
Reviewer 2 Report
The manuscript is now better as a result of corrections made by the authors based on peer review. Congratulations to the authors.